# WGCNA Analysis of Important Modules and Hub Genes of Compound Probiotics Regulating Lipid Metabolism in Heat-Stressed Broilers

**DOI:** 10.3390/ani12192644

**Published:** 2022-10-01

**Authors:** Lihuan Zhang, Xuan Liu, Hao Jia

**Affiliations:** Shanxi Key Lab. for the Modernization of TCVM, College of Life and Science, Shanxi Agricultural University, Jinzhong 030801, China

**Keywords:** compound probiotics, broiler, heat stress, lipid metabolism

## Abstract

**Simple Summary:**

In recent years, the high frequency of high-temperature weather has dramatically increased the possibility of heat stress. Heat stress can affect broilers’ production performance, fat deposition and meat quality through the cells, tissues and organs. The effect of heat stress on fat deposition and meat quality significantly affects the production efficiency of broilers. Probiotics as feed additives can alleviate the adverse reactions of heat stress. Compound probiotics (*Lactobacillus casei*, *Lactobacillus acidophilus* and *Bifidobacterium*) are all colonized in the gut of broilers and are common probiotic supplements. Weighted gene co-expression analysis technology can closely connect the transcriptome with the lipid metabolism results to find the related genes that act on a specific eigenvalue. The purpose of this study was to study the effects of compound probiotics on production performance, meat quality and lipid metabolism in heat-stressed broilers through weighted gene co-expression analysis to determine the genes and modules related to lipid metabolism and to provide a theoretical basis for the poultry industry to alleviate heat stress.

**Abstract:**

This study aimed to study compound probiotics’ (*Lactobacillus casei*, *Lactobacillus acidophilus* and *Bifidobacterium*) effects on production performance, lipid metabolism and meat quality in heat-stressed broilers. A total of 400 one-day-old AA broilers were randomly divided into four groups, each containing the same five replicates, with 20 broilers in each replicate. The control (21 °C) and experiment 2 were fed a basic corn–soybean meal diet. Experiment 1 (21 °C) and experiment 3 were fed a basic corn–soybean meal diet with 10 g/kg compound probiotics on days 7 and 28, respectively. The ambient temperature of experiment 2 and experiment 3 was increased to 30–32 °C (9:00–17:00) for 28–42 days, while the temperature for the other time was kept at 21 °C. The results showed that, compared with the control, the production performance and the content of high-density lipoprotein cholesterol in experiment 1 and triglyceride (TG) in experiment 2 increased (*p* < 0.05). Compared with experiment 2, TG decreased and the production performance increased in experiment 3 (*p* < 0.05). However, there was no significant change in meat quality indicators. Weighted gene co-expression network analysis (WGCNA) was used to analyze the intramuscular fat, abdominal fat and five blood lipid indicators. We found five related modules. Fatty acid biosynthesis, glycerolipid metabolism, and fat digestion and absorption were the pathways for KEGG enrichment. Additionally, *NKX2-1*, *TAS2R40*, *PTH*, *CPB1*, *SLCO1B3*, *GNB3* and *AQP7* may be the hub genes of compound probiotics regulating lipid metabolism in heat-stressed broilers. In conclusion, this study identified the key genes of compound probiotics regulating lipid metabolism and provided a theoretical basis for the poultry breeding industry to alleviate heat stress.

## 1. Introduction

The intensive breeding mode of the poultry industry has met the increasing consumer demand and achieved good economic benefits. However, broilers are vulnerable to many stressors or diseases in this breeding environment [1]. Among them, heat stress is considered one of the most critical stressors in the production process of all types of animals. The changes in ambient temperature and the physiological characteristics of broilers are the main factors causing heat stress. Heat stress can cause changes in the gene, mRNAs, proteins and metabolic levels of broilers and affect the biological functions of cells, tissues, and organs, resulting in increased fat deposition and even death of broilers, thus causing substantial economic losses [2,3]. The intestine is an important organ for broilers to absorb nutrients, which is critical to poultry production and is also a sensitive site for heat stress injury [4]. Although the cecum is not considered the key intestine in other species, it plays a key role in digestion fermentation in broilers, and more information about its molecular function may be important [5]. The effect of heat stress on fat deposition significantly affects the production efficiency of broilers. A certain amount of intramuscular fat (IMF) can enhance meat flavor, juiciness and tenderness and increase consumers’ purchase desire, thus expanding the market needs to affect the production efficiency of broilers. Abdominal fat (AF) is considered useless in the processing of chicken products. Moreover, AF is the main part of fat deposition, and excessive fat deposition is highly related to cardiovascular disease [6]. Currently, the poultry breeding industry has improved the production performance of broilers through genetic breeding and other means. Still, the development of their thermoregulation system does not match the rapid increase in the growth rate, meaning that broilers are unable to adapt to the change in environmental temperature and the addition of the metabolic rate through self-regulation.

Probiotics have been widely used in the production of broilers and have the characteristics of having no residue and non-toxic side effects [7]. In addition, they can regulate nutrient absorption, improve the intestinal barrier and provide a microbial population, as well as play roles in antioxidation, cell apoptosis and immunity, so they can enhance the resistance of broilers to heat stress [7,8]. The effectiveness of probiotics mainly depends on the type and quantity of metabolites produced and secreted. Even in the same species, different probiotics have different roles in preventing or regulating diseases. Therefore, the effect of using a probiotic mixture correctly is better than that of using probiotics alone [8]. *Lactobacillus casei*, *Lactobacillus acidophilus*, and *Bifidobacterium* constitute compound probiotics. They are widely used in the food industry (e.g., in probiotic dairy products and livestock and poultry feed) and are also effectively used in clinical practice. Our previous research found that adding compound probiotics can significantly promote the growth and development of broilers. When the ratio of *L. casei: L. acidophilus: Bifidobacterium* is 1:1:2 and the number is 10 g, the effect is good [9]. Therefore, this study took AA broilers as the research object to explore the impact of compound probiotics on production performance, meat quality and lipid metabolism in heat-stressed broilers and used WGCNA technology to analyze the genes and pathways regulating lipid metabolism in broilers. This study provides a theoretical basis for applying probiotics to improve lipid metabolism in heat-stressed broilers to obtain the maximum production and economic benefits of broilers.

## 2. Materials and Methods

### 2.1. Ethical Statement

This study was strictly approved by the animal care and use Committee of Shanxi Agricultural University (Registration Number, SXAU-EAW-2021C0630).

### 2.2. Feeding Management and Experimental Design

The freeze-dried powder of probiotics (*L. casei*, *L. acidophilus* and *Bifidobacterium*) used in this study was purchased from Shaanxi Ruimao Biotechnology Co., Ltd. *L. casei*, *L. acidophilus* and *Bifidobacterium* were mixed into composite probiotics in the ratio of 1:1:2. The density of 10 g live bacteria added per kilogram of feed is 1 × 10^10^ CFU/g compound probiotics. We bought 400 one-day-old AA broilers (male) from the Xiangfeng Poultry Industry, Taigu County, Shanxi Province. Broilers were vaccinated as required. All broilers were raised in three-layer overlapping stainless-steel vertical cages. The experiment was conducted at the experimental poultry house of the College of Life and Science, Shanxi Agricultural University, China. The room was cleaned and disinfected regularly, and the animals were fed at 8:00 and 19:00 every day and had ad libitum access to water. The room was exposed to light for 23 h and dark for 1 h daily. The brooding temperature was 34–35 °C for 1–3 d and 32–33 °C for 4–7 d, and then it decreased by 1–2 °C every day until 21 °C, maintaining a constant temperature.

A total of 400 one-day-old AA broilers were randomly divided into four groups. One group contained the same five replicates, each replicating with 20 broilers. The feeding period was divided into 1–21 d and 22–42 d. The corn–soybean meal formula was prepared according to the National Research Council recommendations (NRC, 1994) and the Chinese Chicken Feeding Standards (2004). Table 1 shows the nutritional composition of the basic diet. Broilers were fed according to the “Production technique criterion for commercial broilers” (GB/T 19664-2005). The control and experiment 2 were fed a basic corn–soybean meal diet. Experiment 1 and experiment 3 were fed feed containing compound probiotics. Experiment 1 and experiment 3 were fed the feed containing compound probiotics on the 7th and 28th days, respectively. The ambient temperature of the control and experiment 1 was controlled at 21 °C. The ambient temperature of experiment 2 and experiment 3 was increased from 9:00 to 17:00 every day to 30–32 °C for 28–42 days; the heat stress treatment lasted 8 h per day, while the temperature for the other time was kept at 21 °C.

### 2.3. Sample Collection and Processing

The day before slaughter, broilers fasted for 12 h and drank freely. A total of 40 broilers of similar weight were selected from each group for slaughter. The growth data were recorded, and production performance was calculated. The meat quality was measured using the chest muscle. Blood was collected from the carotid artery according to the blood collection procedure [10]. The blood was taken and stored in a 4 °C refrigerator overnight and then centrifuged at 4 °C at 3000 r/min for 20 min, and the serum was stored in a −80 °C refrigerator for determination of blood lipids. We took 2 cm^3^ of the cecum tissue samples and rinsed the contents of the cecum with precooled normal saline. The cecum was frozen in liquid nitrogen and stored in a refrigerator at −80 °C for transcriptome sequencing.

### 2.4. Production Performance

On day 42, the body weight, dressed weight, half-eviscerated weight and eviscerated weight of broilers in each group were recorded. During the experiment, the feed intake was recorded regularly. Average daily gain (ADG), average daily feed intake (ADFI) and feed conversion ratio (FCR) were calculated at the end of the test. The production performance indexes were measured according to the Performance Terms and Measurements for Poultry (NY/T 823-2004).

### 2.5. Meat Quality Test

(1)Dripping loss: We took the chest muscle, weighed it as W_1_, tied one end of the meat sample with a thin wire, sealed it and hung it in a 4 °C refrigerator. After 24 h, the meat sample was wiped with dry filter paper and weighed as W_2_, and dripping lose was calculated according to the formula Dripping loss = [(W_1_ − W_2_)/W_1_] × 100%.(2)Crude moisture: We took the chest muscle and determined crude moisture through direct drying according to GB 5009.3—2016 National food safety standard Determination of moisture in foods.(3)Cooking loss: We took out the chest muscle sample stored in a 4 °C refrigerator for 24 h, making the sample temperature consistent with the room temperature, peeled off the surrounding fat and broken meat and weighed it as M_1_. We placed the sample in a plastic bag. Finally, the sample was placed in a 75 °C water bath for 45 min, removed and cooled to room temperature, wiped with filter paper and weighed as M_2_. Cooking loss was calculated according to the formula Cooking loss = [(M_1_ − M_2_)/M_1_] × 100%.(4)Cooked meat rate: We weighed the chest muscle sample as S_1_, steamed it in a pot of boiling water for 30 min, took out the sample, cooled it for 15 min and dried it with filter paper. It was weighed as S_2_ and the cooked meat rate was calculated according to the formula Cooked meat rate = (S_2_/S_1_) × 100%.(5)Shear force: We took the chest muscle according to the national standard NY_T 1180—2006 Determination of meat tenderness Shear force method.(6)Intramuscular fat: We took the chest muscle and measured it according to GB 5009.6—2016 National food safety standard Determination of fat in food.(7)The abdominal fat rate was calculated using the following formula: AF rate = AF weight/(eviscerated weight + AF weight) × 100%. The eviscerated weight was determined according to the national standard NY/T 823-2004 Performance Terms and Measurements for Poultry.

### 2.6. Blood Lipid Test

The contents of triglyceride (TG), total cholesterol (TC), high-density lipoprotein cholesterol (HDL-C) and low-density lipoprotein cholesterol (LDL-C) in broiler serum were determined using an automatic biochemical analyzer.

### 2.7. WGCNA Analysis

#### 2.7.1. Construction of Gene Co-Expression Network

Four cecal samples were randomly selected from each group and sent to Shanghai Meiji Biomedical Technology Co., Ltd. (Shanghai, China) for transcriptome sequencing. The gene expression information of the sequencing results (Appendix A) and lipid phenotype data were used for weighted gene co-expression network analysis (WGCNA) using the WGCNA package in the RStudio application (Version 4.1.2). Firstly, the picksoftthreshold function was used to determine the optimal soft threshold so that the correlation intensity between genes conforms to the scale-free distribution and the scale-free network maintains a certain connectivity. The mergecutheight of similar modules was set to 0.25, the Tom type was unsigned and the minimum number of genes constituting the module (minmodulesize) was 30 to obtain the gene module after the construction of the co-expression network.

#### 2.7.2. Screening Key Modules and Hub Genes

The phenotype data related to lipid metabolism (IMF, AF, TG, TC, HDL-C and LDL-C) and gene modules were analyzed using Pearson correlation. The modules with correlation coefficient |r| ≥ 0.56 and *p* < 0.05 were selected as key modules for subsequent analysis. Using the RStudio application to analyze the KEGG enrichment of key module genes, when *p* < 0.05, this indicated that the KEGG signal pathway was significantly enriched. At the same time, the correlation coefficient between each gene in the module and the phenotype data were calculated to obtain the gene significance (GS), and the MM-GS scatter diagram was obtained by combining the correlation between the gene and its module eigenvector (MM). The higher the gene connectivity in the module, the more the gene is at the network’s core. Therefore, the genes with GS > 0.4 and |MM| > 0.8 were screened for analysis. Using the software Cytoscape (Version 3.8.2), the top 30 genes in terms of the MCC value were screened and displayed as hub genes.

### 2.8. Statistical Analysis

The original data were summarized in Excel and analyzed using SPSS (IBM SPSS Statistics 26). One-way ANOVA and the post hoc Duncan multiple range test were used to compare the groups’ differences. *p* < 0.05 indicates a significant difference. All data are presented as the mean ± SEM.

## 3. Results

### 3.1. Production Performance

It can be seen from Table 2 that compared with the control group, the body weight, ADG, dressed weight, eviscerated weight and half-eviscerated weight of experiment 1 increased, while the FCR decreased (*p* < 0.05). On the contrary, the changes in experiment 2 were opposite to those in experiment 1 (*p* < 0.05). The body weight, ADG, dressed weight, eviscerated weight and half-eviscerated weight of experiment 3 were higher than those of experiment 2, but the FCR was lower (*p* < 0.05).

### 3.2. Meat Quality

It can be seen from Table 3 that compared with the control, the dripping loss, cooking loss, shear force and AF rate in experiment 1 decreased, while the crude water content, cooked meat rate and IMF increased. On the contrary, the dripping loss, cooking loss and shear force in experiment 2 increased, while the crude water content, cooked meat rate and IMF decreased. Compared with experiment 2, the dripping loss, cooking loss, shear force and AF rate of broilers in experiment 3 decreased, while the crude moisture, cooked meat rate and IMF increased (*p* > 0.05). 

### 3.3. Blood Lipid Quality

Table 4 shows that, compared with the control group, the HDL-C content of broilers in experiment 1 increased significantly (*p* < 0.05), while the TG, TC, and LDL-C contents decreased, but the difference was not significant. Additionally, compared with the control, the content of TG increased significantly (*p* < 0.05) and the content of HDL-C decreased in experiment 2. Compared with experiment 2, TG in experiment 3 was significantly lower and the HDL-C content was significantly higher (*p* < 0.05). 

### 3.4. WGCNA Analysis

#### 3.4.1. Construction and Module Division of Gene Co-Expression Network

Lipid metabolism phenotype data (IMF, AF, TG, TC, HDL-C and LDL-C) and transcriptome data were analyzed jointly. WGCNA analysis can further explore broilers’ potential hub genes regulating lipid metabolism. As shown in Figure 1, when the soft power was 8, the fitting index R^2^ of the scale-free network was greater than 0.8. The Tom value was used to carry out hierarchical clustering, and the dynamic mixed clipping method was used to identify the co-expression modules. A total 26 modules were obtained, and each module corresponded to a color (Figure 2, Appendix A). The exact number of genes is shown in Figure 3. The number of genes in the turquoise module was the largest, with a total of 4160. The smallest module was the orange, with a total of 64. Furthermore, the number of genes in other modules was between 64 and 4160. 

#### 3.4.2. Correlation between Gene Module and Phenotype Data

According to the correlation between modules and IMF, AF, TG, TC, HDL-C and LDL-C (Figure 4), five modules were selected based on |R| ≥ 0.56 and *p* < 0.05. Appendix A shows the correlation and *p*-value between the module and the phenotype. Among them, the cyan module was positively correlated with IMF (404 genes), the brown module was positively correlated with TG (1366 genes), the grey60 module was positively correlated with HDL-C (328 genes), the orange module was positively correlated with LDL-C (64 genes) and the magenta module was negatively correlated with TC (744 genes). 

#### 3.4.3. Enrichment Analysis of Module Gene KEGG

In this study, KEGG enrichment analysis was carried out for the genes in the five selected key modules, as shown in Figure 5 and Appendix A, showing the first 20 significantly enriched pathways. The cyan module was enriched with 178 KEGG pathways, of which 12 pathways were significantly enriched, such as pyruvate metabolism and fatty acid biosynthesis. A total 311 KEGG pathways were enriched in the brown module, of which 33 pathways were significantly enriched. For example, the significant modules included arrhythmogenic right ventricular cardiomyopathy, hypertrophic cardiomyopathy, and insulin secretion. All 275 KEGG pathways were enriched in the magenta module, of which 34 pathways were significantly enriched, including phospholipase fat digestion and absorption, D signaling pathway and glycerolipid metabolism. There were 170 KEGG pathways in the grey60 module, of which 17 were significantly enriched, including vitamin digestion and absorption, non-alcoholic fatty liver disease and mineral absorption. There were 79 KEGG pathways in the orange module, of which 8 pathways were significantly enriched, including the Ras signaling pathway, MAPK signaling pathway and glutamatergic synapse.

#### 3.4.4. Hub Gene Screening of Lipid Metabolism-Related Modules

We plotted the scatter diagram of each module’s GS value and MM value (Figure 6). We set GS > 0.4 and |MM| > 0.8 to screen key genes. We obtained 122, 264, 178, 89 and 32 hub genes in the cyan module, brown module, magenta module, grey60 module and orange module, respectively. The cytohubba plug-in of the Cytoscape software was used to screen the top 30 hub genes according to the MCC value (Figure 7, Appendix A). The closer the color was to red, the higher the MCC value. The higher the MCC value, the higher the correlation with the phenotype.

## 4. Discussion

Heat stress can reduce the growth rate and immune capacity of broilers, resulting in poor meat quality and reduced welfare [11]. Economic analysis shows that the domestic poultry industry will lose USD 128 million annually due to heat stress [12]. Production performance is an important index to evaluate whether poultry production efficiency is in line with people’s ideals. The research shows that heat stress can induce intestinal injury, immunosuppression and oxidative stress to adversely affect growth performance by reducing the feed intake and feed utilization rate of broilers [13]. In addition, heat stress can also damage carcass traits by affecting energy metabolism and redox status, leading to a decline in broiler meat yield [14]. Studies have found that adding a probiotic complex to the diet could alleviate the decline in broiler performance caused by heat stress [15,16]. Moustafa et al. showed that adding 1.5% spirulina to the feed could improve heat stress and reduce the carcass level of broilers [17]. This is consistent with our results. This study showed that heat stress reduced the body weight, ADG, eviscerated weight and half-eviscerated weight of broilers and increased the FCR. However, the supplement of compound probiotics improved the performance of broilers. This effect of adding probiotics to improve the growth performance of heat-stressed broilers may be due to the colonization of probiotics in the intestinal tract, which increases the activity of digestive enzymes, promotes the fermentation and digestion of nutrients and energy absorption and produces growth-promoting factors, thus stimulating intestinal peristalsis and improving feed digestibility and utilization [18].

Birds exposed to a high temperature will change their physiological homeostasis, which may induce systemic immune disorders and/or endocrine and electrolyte disorders, resulting in heat stress injury [13]. Heat stress can negatively affect the meat quality of broilers by changing physiological processes such as aerobic metabolism, glycolysis and fat deposition [1]. Previous studies have shown that adding probiotics (*Bacillus subtilis*) to the diet can improve the yield and quality of poultry products and reduce the pressure caused by heat stress [19]. Many studies have shown that probiotic supplements can reduce the percentage of AF in broilers [20,21,22]. Adding *Bacillus subtilis* to the diet can reduce the shear force, significantly reduce the cooking loss and dripping loss of broiler breast muscle and significantly improve the water-holding capacity of broiler breast muscle [23,24,25]. In addition, in a high-temperature environment, heat stress can increase the dripping loss and cooking loss of meat quality in broilers, which may be due to the oxidative stress caused by heat stress damaging the function of the cell membrane, resulting in increased permeability [26]. Still, the above damage can be effectively improved by supplementing the diet with 1.5 g/kg compound synbiotic [27]. Dietary supplements of selenium-rich probiotics can also significantly increase broiler breast muscle’s water-holding capacity and tenderness to improve meat quality [28]. In addition, *Clostridium butyricum* as a feed supplement can significantly increase the content of IMF in broilers [29]. This study found that the compound probiotics reduced the dripping loss, cooking loss, shear force and AF rate of heat-stressed broilers and increased the crude water content, cooked meat rate, and IMF. No significant effect could be attributed to the chicken varieties used in the experiment, the different heat stress treatments or the bacterial species. The above results indicate that compound probiotics may reduce the muscle water loss rate by maintaining the integrity of the cell membrane, regulating the body’s lipid metabolism, increasing the IMF content and reducing the deposition of AF in poultry by inhibiting lipid biosynthesis and promoting the decomposition and metabolism of fatty acids. Therefore, compound probiotics can improve the meat quality of heat-stressed broilers and alleviate heat stress injury. 

Heat stress can lead to liver damage, lipid metabolism disorders and excessive fat deposition in broilers [26], and excessive fat deposition usually leads to hypercholesterolemia and atherosclerosis. Methods such as drug interference and feed additives inhibit excessive fat deposition in broilers. The blood lipid content is an important indicator reflecting fat deposition in broilers. Among them, HDL-C is involved in reverse cholesterol transport, helping to extract excess cholesterol deposited in blood vessel walls and send it back to the liver, which is excreted through the gastrointestinal tract [30]. HDL-C has anti-inflammatory and antioxidant effects, can inhibit the oxidation of harmful bacteria and phospholipids in the intestine and can reduce LDL-C activity. Lower LDL-C and higher HDL-C help maintain vasodilation, thereby promoting better blood flow [31]. The increase in TG, TC and LDL-C concentrations is an essential factor leading to cardiovascular disease [32], and decreasing the concentration can also improve the quality of meat [33]. Studies have shown that heat stress can increase the content of TG, TC and LDL-C and reduce the HDL-C content in broilers [34]. However, Yazhini et al. [35] showed that encapsulated probiotics (*Lactobacillus* and *Bifidobacterium*) could reduce broiler serum’s TC and TG content in a typical production environment. *Lactobacillus* can reduce the TC content by assimilating intestinal endogenous or exogenous TC and can also reduce TC absorption by reducing or inhibiting the expression level of Niemann–Pick C1-like 1 protein expressed on the surface of intestinal cells [36,37]. In addition, *Bifidobacterium* can also reduce the TC content in the body through assimilation, co-precipitation, adsorption and binding [38]. Panda et al. [39] found that dietary supplementation of *Bacillus* (100 mg/kg feed, with 6 × 10^8^ spores per g) could reduce the levels of LDL-C and TG in the serum of broilers. Under heat stress, the lipolysis rate of broilers decreases, and fat deposition increases. Cr can affect the synthesis and decomposition of fat and participate in the metabolism of carbohydrates; Cr supplementation can significantly reduce broilers’ serum TG and LDL-C and increase HDL-C [40]. The content of HDL-C has a positive effect on the lipid metabolism of broilers [41]. Adding 0.1% betaine supplementation to a basal diet also significantly reversed the adverse effects of elevated LDL-C content in heat-stressed broilers, thereby improving carcass formation by altering lipid metabolism [42]. In this experiment, compound probiotics reduced TG and LDL-C in broilers and significantly increased HDL-C, which indicates that the compound probiotics had an inhibitory effect on fat deposition in broilers and reduced the occurrence of cardiovascular disease. The compound probiotics did not reduce the TC of the heat-stressed broilers, which may have been caused by the severe damage to the heat-stressed broilers.

Lipid metabolism is a complex process regulated by multiple genes and regulators through different signaling pathways. Currently, most studies only focus on finding differentially expressed genes but ignore the association between genes and phenotypes. In this study, a gene co-expression network was constructed to further explore the key genes regulating lipid metabolism in broilers at an overall level. The phenotypic data IMF, AF, TG, TC, HDL-C, and LDL-C were analyzed using WGCNA technology and transcriptome results, and five related modules were identified. The cyan, brown, magenta, grey60 and orange modules were closely related to IMF, TG, TC, LDL-C, and LDL-C, respectively. These modules contained 404, 1366, 744, 328 and 64 genes. KEGG enrichment analysis of genes in each module showed that the pathways of fatty acid biosynthesis and pyruvate metabolism were related to the regulation of IMF; insulin secretion and hypertrophic cardiomyopathy were related to the regulation of TG; glycerolipid metabolism, insulin secretion and fat digestion and absorption were related to the regulation of TC; and mineral absorption, non-alcoholic fatty liver disease, and vitamin digestion and absorption were related to the regulation of HDL-C. Moreover, the Ras signaling pathway and other pathways were related to LDL-C regulation. Both fatty acids and pyruvate are involved in the formation of fat, which may promote the formation of intramuscular fat. Fat accumulation is a complex process related to TG and TC metabolism [43]. In addition, insulin can regulate the secretion of hepatic TG by stimulating adipose tissue lipoprotein lipase (LPL) [44]. Vitamin D and minerals can regulate HDL-C. Vitamin D deficiency will lead to HDL-C dyslipidemia [45]. At the same time, dietary zinc supplementation can improve the HDL-C level of broilers [46]. Long-term intake of a high-fat diet can increase the LDL-C content, which can lead to miRNA imbalance, the Ras signaling pathway is involved in the regulation of dyslipidemia [47].

The GS, MM and MCC values were further used to screen out each module’s top 30 hub genes. For example, *IP6K3* and *NKX2-1* were the hub genes in the cyan module. Chatree et al. [48] showed that *IP6K3* is associated with obesity and can promote insulin cycling and reduce glucose. Upregulation of *NKX2-1* (Thyroid transcription factor-1) can reduce the accumulation of AF in chickens [49]. The AF of chickens is considered to be a by-product with very low economic value, and the reduced blood sugar can be converted into fat, thus promoting the deposition of IMF [50]. *TAS2R40* was the hub gene in the brown module. *TAS2R40* expressed in enteroendocrine cells can reduce the TG content by secreting incretin [51]. Moreover, *PTH* was the core gene in the magenta module. Studies have shown that *PTH* stimulates the synthesis of vitamin D_3_, inhibiting the synthesis of endogenous cholesterol [52,53]. Vitamin D deficiency can increase the level of *PTH*, which can increase the risk of cardiovascular diseases such as myocardial hypertrophy and valve calcification by influencing lipid metabolism and increasing inflammation, insulin resistance, hypertension and other mechanisms [54]. A higher level of vitamin D can increase intestinal calcium absorption, inhibit the absorption of cholesterol, promote the excretion of cholesterol from feces, and reduce the formation of TG [55]. *TAFA5*, *CPB1*, and *SLCO1B3* were the hub genes in the grey60 module. *TAFA5*, also known as *FAM19A5*, plays a protective role in atherosclerosis, obesity and inflammation [56]. As a new fat protective factor, *TAFA5* can inhibit the proliferation, migration and neointima formation of vascular smooth muscle cells after injury [57]. *CPB1* is Carboxypeptidase B1, which is involved in protein hydrolysis and is considered related to chicken feed digestion, so *CPB1* may be involved in the decomposition of protein in HDL-C [58]. Furthermore, in humans, *CPB1* is thought to be involved in lipid storage and lipid droplet formation [59]; in mice, *CPB1* is associated with insulin activation [60]. *SLCO1B3* is a member of the solute carrier organic anion transporter family and is an important membrane lipid transporter, which can regulate hepatic fatty acid metabolism by participating in fat metabolism and exogenous substance transport, indirect transport of thyroid hormones and estrogen, and bile acid transport [61,62,63]. Genes such as *GNB3* and *AQP7* were the hub genes in the orange module. *GNB3* is associated with obesity, and enhanced *GNB3* signaling increases the risk of brain disease, obesity, hypertension and coronary heart disease [64]. In addition, *GNB3* overexpression leads to obesity and glucose intolerance in mice and produces a series of metabolic syndromes such as acute thermogenic dysregulation [65]. Cardiovascular disease and obesity usually show an increase in the LDL-C content [66]. The aquaglycerol channel protein AQP7 is involved in gluconeogenesis in broiler chickens and insulin secretion and triacylglycerol synthesis in mice [5]. Furthermore, *AQP7,* as an important channel for glycerol outflow from adipocytes, plays a key role in regulating TG accumulation, glucose homeostasis, fat transporters and fat accumulation [67]. Therefore, we inferred that compound probiotics may regulate lipid metabolism in heat-stressed broilers by regulating fatty acid biosynthesis, pyruvate metabolism, insulin secretion, glycerolipid metabolism, fat digestion and absorption, mineral absorption, vitamin digestion and absorption and the *NKX2-1*, *TAS2R40*, *PTH*, *CPB1*, *SLCO1B3*, *GNB3* and *AQP7* genes.

## 5. Conclusions

In conclusion, compound probiotics increased the production performance and HDL-C and significantly reduced TG in heat-stressed broilers. They also improved blood lipid levels, specifically through fatty acid biosynthesis, pyruvate metabolism, insulin secretion, glycerolipid metabolism, fat digestion and absorption, mineral absorption, and vitamin digestion and absorption, as well as *NKX2-1*, *TAS2R40*, *PTH*, *CPB1*, *SLCO1B3*, *GNB3*, *AQP7*, and other genes that regulate lipid metabolism in heat-stressed broilers, improving meat quality and alleviating heat stress injury. This confirmed the key gene of compound probiotics regulating lipid metabolism and provided a theoretical basis for poultry breeding industry to alleviate heat stress.

## Figures and Tables

**Figure 1 animals-12-02644-f001:**
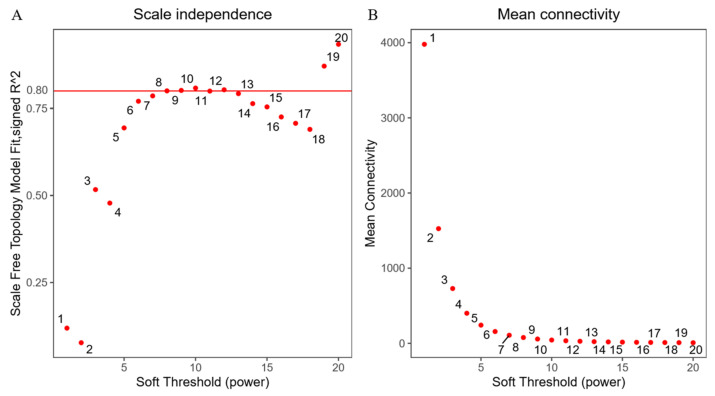
Determination of soft threshold. (**A**) Ordinate indicates correspondence between β Value and the goodness of fit R^2^ between the adjacency matrix after transforming the scale-free network assumption. When the soft threshold is 8, R^2^ is greater than 0.8. (**B**) The ordinate represents the average connectivity of each node in the network, determined by the corresponding β the adjacency matrix representation after value conversion.

**Figure 2 animals-12-02644-f002:**
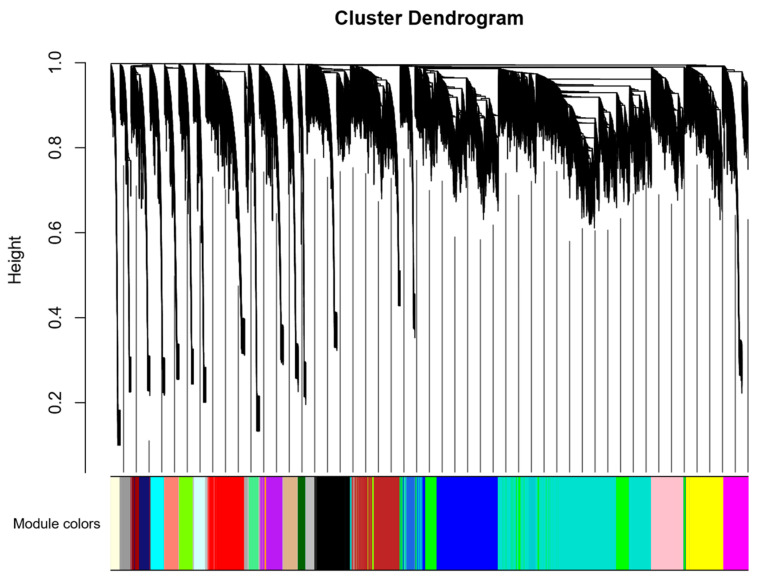
Module classification tree. This figure reflects the clustering of genes within the module.

**Figure 3 animals-12-02644-f003:**
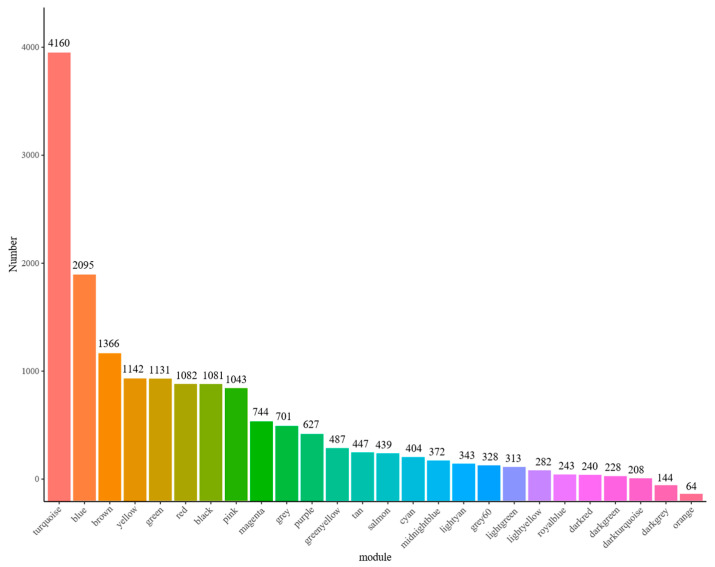
Number distribution of module genes. The abscissa represents the name of the module, and the ordinate represents the number of genes.

**Figure 4 animals-12-02644-f004:**
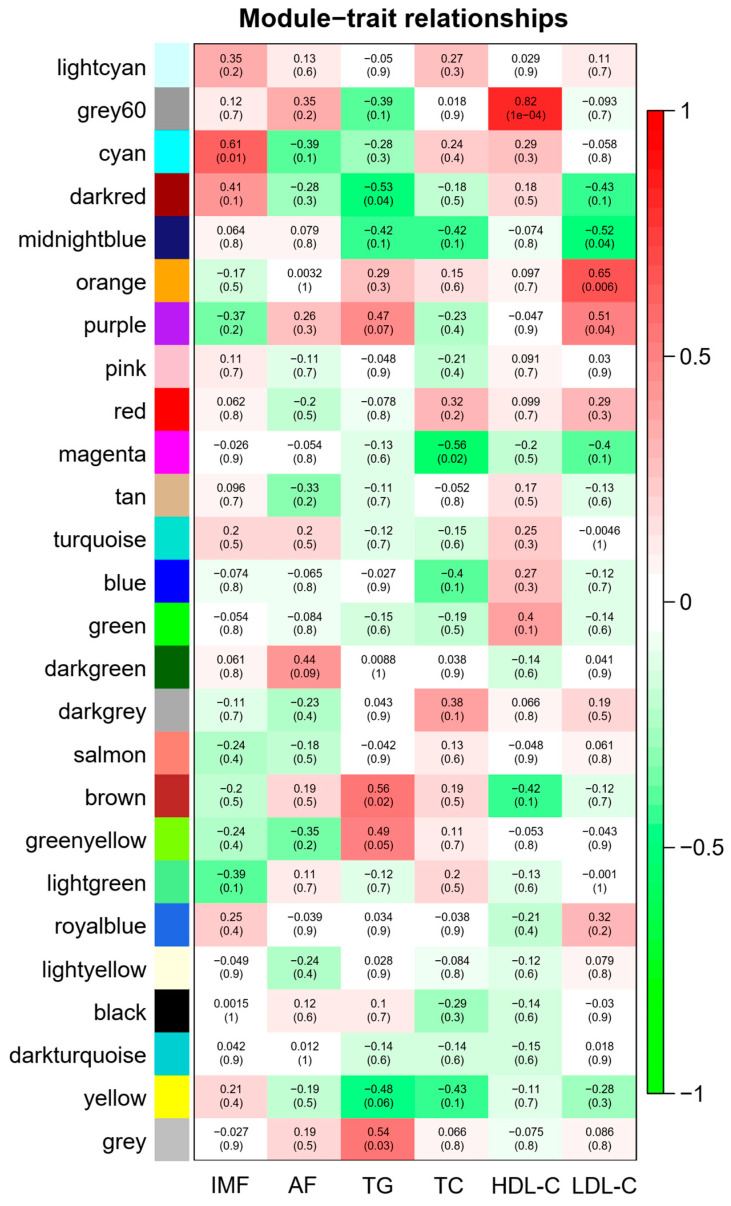
Heat map of correlation between modules and phenotypic data. IMF (intramuscular fat), AF (abdominal fat), TG (triglyceride), TC (total cholesterol), HDL-C (high-density lipoprotein cholesterol), LDL-C (low-density lipoprotein cholesterol). The legend on the right represents the correlation between phenotype and module. The numbers in brackets represent the *p*-value between the module and the phenotype.

**Figure 5 animals-12-02644-f005:**
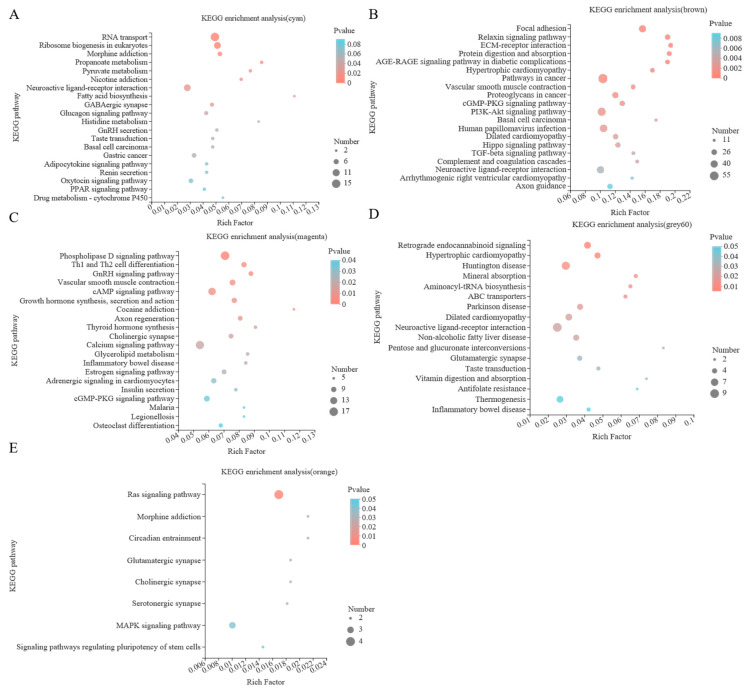
Scatter diagram of KEGG enrichment analysis. (**A**) cyan module, (**B**) brown module, (**C**) magenta module, (**D**) grey60 module, and (**E**) orange module.

**Figure 6 animals-12-02644-f006:**
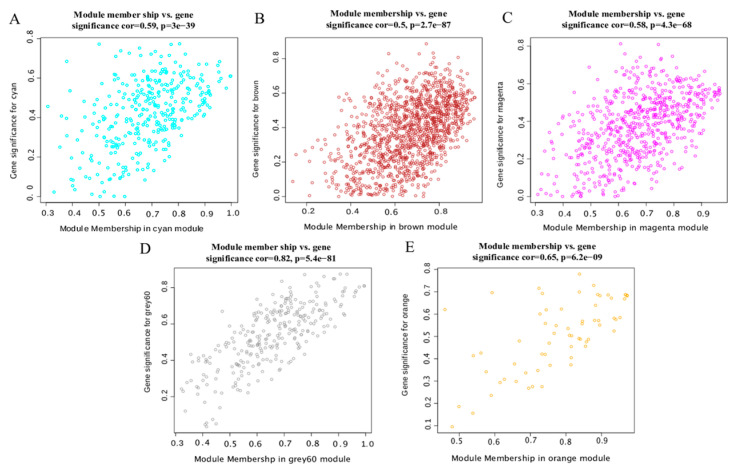
Scatter plot of GS and MM. (**A**) cyan module, (**B**) brown module, (**C**) magenta module, (**D**) grey 60 module, and (**E**) orange module.

**Figure 7 animals-12-02644-f007:**
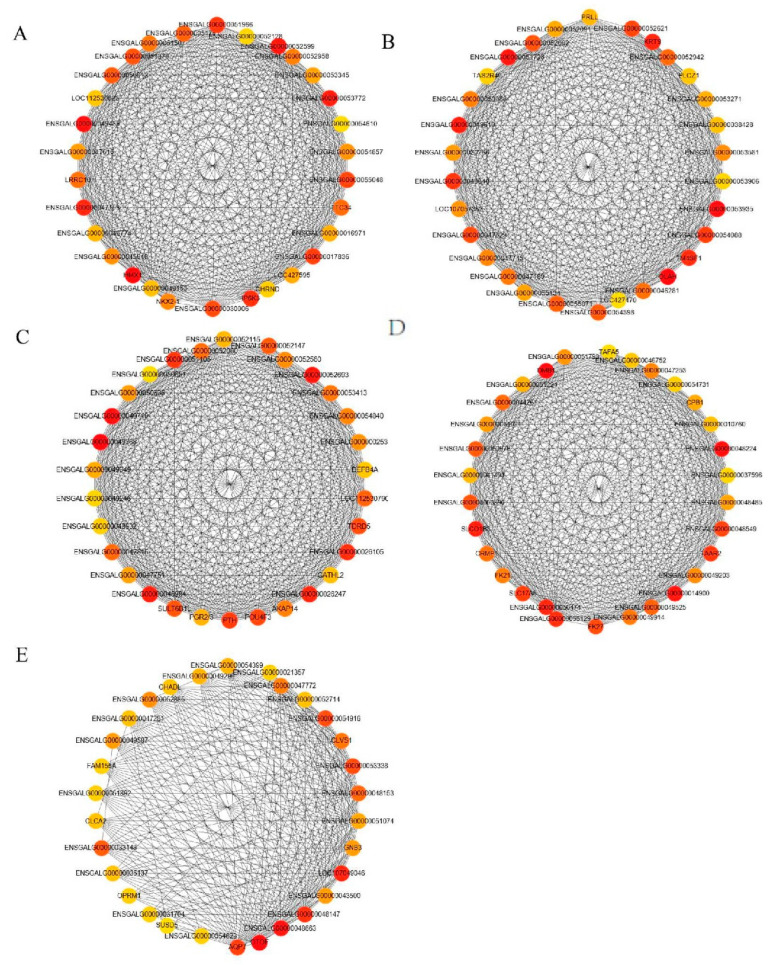
Gene interaction network diagram of the top 30 MCC value in the module. (**A**) cyan module, (**B**) brown module, (**C**) magenta module, (**D**) grey60 module, and (**E**) orange module.

**Table 1 animals-12-02644-t001:** Composition of and nutrient levels in the basic diet.

Item	1–21 Days	22–42 Days
Diet composition, %		
Corn	56.49	61.42
Soybean oil	2.22	3.00
Soybean meal	30.24	25.30
Cotton seed meal	5.00	5.00
Fishmeal	2.43	1.98
CaHCO_3_	1.60	1.39
Limestone	1.16	1.10
Methionine	0.15	0.05
NaCl	0.30	0.35
Choline	0.19	0.19
Premix ^1^	0.22	0.22
Nutrient, % ^2^		
ME (MJ·kg^−1^)	12.12	12.54
Crude protein	21.00	19.00
Lysine	1.12	0.98
Methionine + Cystine	0.84	0.68
Calcium	1.00	0.90
Available phosphorus	0.30	0.30

^1^ The premix contained 0.2% trace elements and provided the following nutrients per kg feed: Fe 80 mg, Mn 80 mg, Zn 80 mg, I 0.35 mg, Se 0.15 mg. The premix contained 0.02% vitamins per kg feed: vitamin D_3_ 3000.00 IU, vitamin E 30.00 IU, vitamin K_3_ 1.00 mg, vitamin B_1_ 2.00 mg, vitamin B_2_ 6.00 mg, Pantothenic acid 9.00 mg, Pyridoxine 5.00 mg, Niacin 30.00 mg, vitamin B_12_ 0.01 mg, Biotin 0.10 mg, Folic acid 0.30 mg. ^2^ Nutrition level is a calculated value.

**Table 2 animals-12-02644-t002:** Production performance of broiler.

Item ^1^	Control	Experiment 1	Experiment 2	Experiment 3	*p*-Value
Body Weight, g	2911.243 ± 10.092 ^b^	3121.163 ± 24.104 ^a^	2658.173 ± 39.407 ^c^	2894.857 ± 82.152 ^b^	0.001
ADG, g/brid	89.010 ± 0.967 ^b^	106.317 ± 1.971 ^a^	73.463 ± 5.239 ^c^	92.624 ± 4.528 ^b^	0.002
ADFI, g/brid	168.630 ± 1.649	168.780 ± 3.296	151.144 ± 2.975	167.542 ± 15.013	0.372
FCR, g/g	1.895 ± 0.039 ^ab^	1.590 ± 0.059 ^c^	2.073 ± 0.105 ^a^	1.802 ± 0.077 ^bc^	0.011
Dressed Weight, g	2436.667 ± 46.369 ^ab^	2656.333 ± 24.694 ^a^	2297.667 ± 58.373 ^b^	2622.000 ± 161.970 ^a^	0.071
Eviscerated Weight, g	1899.333 ± 52.922 ^ab^	2008.000 ± 58.141 ^a^	1835.667 ± 28.109 ^b^	1988.000 ± 47.438 ^ab^	0.110
Half-eviscerated Weight, g	2153.787 ± 18.068 ^b^	2598.653 ± 188.262 ^a^	2038.098 ± 16.619 ^b^	2420.005 ± 159.501 ^ab^	0.047

^1^ Control, Broilers were fed a basic diet and raised at 21 °C; Experiment 1, seven-day-old broilers were fed a basic diet containing 10 g compound probiotics/kg and raised at 21 °C; Experiment 2, Broilers were fed a basic diet and raised at 30–32 °C for 28–42 d, while the temperature for the other time kept at 21 °C; Experiment 3, twenty-eight days old broilers were fed a basic diet containing 10 g compound probiotics/kg and raised at 30–32 °C for 28–42 d, while the temperature for the other time kept at 21 °C. The compound probiotics are composed of *Lactobacillus casei*, *Lactobacillus acidophilus* and *Bifidobacterium* in the ratio of 1:1:2, and the density of living bacteria is 1 × 10^10^ CFU/g. Data are presented as the mean ± SEM; ADG, average daily gain; ADFI, average daily feed intake; FCR, feed conversion ratio. ^a,b,c^ Values in the same row with different letter superscripts mean significant differences (*p* < 0.05). *p*-Value represents the significance between groups, different superscript letters in the same line represent significant differences (*p* < 0.05).

**Table 3 animals-12-02644-t003:** Meat quality of broiler.

Item ^1^	Control	Experiment 1	Experiment 2	Experiment 3	*p*-Value
Dripping loss, %	4.352 ± 0.274	3.866 ± 0.532	4.614 ± 0.287	4.254 ± 0.315	0.571
Crude moisture, %	74.939 ± 0.251	75.479 ± 0.802	74.441 ± 0.443	74.741 ± 0.644	0.644
Cooking loss, %	11.840 ± 0.817	11.190 ± 0.418	12.243 ± 0.392	11.248 ± 0.822	0.629
Cooked meat rate, %	62.112 ± 0.488	63.095 ± 0.318	61.936 ± 0.398	62.563 ± 0.444	0.277
Shear force, N	25.783 ± 1.444	23.208 ± 0.848	26.132 ± 1.276	24.611 ± 1.421	0.410
IMF, %	2.577 ± 0.430	2.978 ± 0.521	2.355 ± 0.550	3.669 ± 0.557	0.336
AF rate, %	2.333 ± 0.229	1.978 ± 0.152	2.163 ± 0.270	2.114 ± 0.297	0.781

^1^ Control, Broilers were fed a basic diet and raised at 21 °C; Experiment 1, seven-day-old broilers were fed basic diet containing 10 g compound probiotics/kg and raised at 21 °C; Experiment 2, Broilers were fed a basic diet and raised at 30–32 °C for 28–42 d, while the temperature for the other time kept at 21 °C; Experiment 3, twenty-eight days old broilers were fed a basic diet containing 10 g compound probiotics/kg and raised at 30–32 °C for 28–42 d, while the temperature for the other time kept at 21 °C. The compound probiotics are composed of *Lactobacillus casei*, *Lactobacillus acidophilus* and *Bifidobacterium* in the ratio of 1:1:2, and the density of living bacteria is 1 × 10^10^ CFU/g. Data are presented as the mean ± SEM; IMF, intramuscular fat; AF, abdominal fat. *p*-Value represents the significance between groups, different superscript letters in the same line represent significant differences (*p* < 0.05).

**Table 4 animals-12-02644-t004:** Blood lipid indexes of broiler.

Item ^1^	Control group	Experiment 1	Experiment 2	Experiment 3	*p*-Value
TG, mmol/L	0.325 ± 0.008 ^b^	0.317 ± 0.006 ^b^	0.428 ± 0.028 ^a^	0.257 ± 0.022 ^c^	0.001
TC, mmol/L	4.213 ± 0.107	4.163 ± 0.197	4.236 ± 0.108	4.289 ± 0.201	0.955
HDL-C, mmol/L	2.751 ± 0.019 ^b^	2.846 ± 0.096 ^ab^	2.698 ± 0.075 ^b^	3.171 ± 0.207 ^a^	0.067
LDL-C, mmol/L	1.387 ± 0.049	1.280 ± 0.140	1.545 ± 0.191	1.310 ± 0.212	0.659

^1^ Control, Broilers were fed a basic diet and raised at 21 °C; Experiment 1, seven-day-old broilers were fed basic diet containing 10 g compound probiotics/kg and raised at 21 °C; Experiment 2, Broilers were fed basic diet and raised at 30–32 °C for 28–42 d, while the temperature for the other time kept at 21 °C; Experiment 3, twenty-eight days old broilers were fed basic diet containing 10 g compound probiotics /kg and raised at 30–32 °C for 28–42 d, while the temperature for the other time kept at 21 °C. The compound probiotics are composed of *Lactobacillus casei*, *Lactobacillus acidophilus* and *Bifidobacterium* in the ratio of 1:1:2, and the density of living bacteria is 1 × 10^10^ CFU/g. Data are presented as the mean ± SEM; TG, triglyceride; TC, total cholesterol; HDL-C, high-density lipoprotein cholesterol; LDL-C, low-density lipoprotein cholesterol. ^a,b,c^ Values in the same row with different letter superscripts mean significant differences (*p* < 0.05). *p*-Value represents the significance between groups, different superscript letters in the same line represent significant differences (*p* < 0.05).

## Data Availability

The data presented in this study are available in Appendix A here.

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
