# Peer review of "WGCNA Analysis of Important Modules and Hub Genes of Compound Probiotics Regulating Lipid Metabolism in Heat-Stressed Broilers"

_animals, 2022, doi:10.3390/ani12192644_

Round 1

Reviewer 1 Report

1.      English symbols not correct used, including ℃, roman symbol and bracket. You must correct all the mistake.

2.      The format of the three-line table is not standardized, please check and correct. Table use need the unified and correct form.

3.      Poorly resolution of Figure 5 and Figure 7 need to be replaced or adjusted.

4.      Transcriptome sequencing data is abundant, you need provide the result. Also, it is will be better if you can describe more details related to production performance.

5.      The discussion section needs to be further discussed instead of listed data only, including pathway enrichment or focused genes.

6.      Which phenotypes show that 30-32 ℃ is the optimal temperature for heat stress?Why selected 30-32 ℃ as the optimum temperature for heat stress?

7.      Some required descriptions should be clearly written in the experimental design. For example, what are the feeding standards based on? What standard is the feed formula based on?

8.      What is the standard for the number of probiotics added?

9.      The minimum standard for R2 is 0.8. It is suggested that 0.8 can be marked on the Figure1A.

10.   How to control the temperature difference between the three layers when broiler chickens are raised three-layer overlapping stainless steel vertical cages?

11.   The naming of "jiyinbiaodaliang" is not standardized in the Table S1. What’s meaning “6-4    6-3,6-2,6-1,7-4,7-3,7-2,7-1”, How to measure the level of gene expression, FPKM or readscount? Not described in Table S1.

12.   Table S2's header is wrong, please check the headers of all tables.

13.   There is pvalue, padj, foldchange and up/down-regulated information in gene expression, please complete.

14.   Please proofread the information throughout the article, especially the attached table has many errors.

15.   Broilers were given heat stress on day 28, why were they fed compound probiotics diet on day 7?

Author Response

Dear Reviewer 1:

On behalf of my co-authors, we thank you very much for giving us an opportunity to revise our manuscript, we appreciate you very much for their positive and constructive comments and suggestions on our manuscript entitled “animals-1873223”. 
We carefully studied the comments made by the reviewers and revised them. We provide a revised version for your review.In addition, we recruited native English speakers to revise our articles. Attached please find the responses to the comments from Reviewer 1, which we would like to submit for your kind consideration. 

We would like to express our great appreciation to you and reviewers for comments on our paper. Looking forward to hearing from you.
Thank you and best regards.

Yours sincerely,
Mrs. Zhang
Corresponding author:
Name: Lihuan Zhang
E-mail: [email protected]

Reviewer 2 Report

I strongly advise you to ask a native English speaker to revise your manuscript. You did a job in designing and executing the study, however, you need to improve on the grammar throughout the whole manuscript. See the rest of the comments below:

Line 12-13: The statement, “Excessive fat deposition is highly 12 related to human cardiovascular disease.” Looks out of place.

Line 25: Define what basic diet is when introducing it. For example, “Experiment 1 and experiment 3 were fed a corn-soybean meal basic diet……” The reader needs to briefly understand what that diet was in the abstract.

Line 42: stress reactions??

Line 48: Growth speed? You could probably use growth rate

Line 51-53. This sentence could be restructured differently. Re-write it without starting it with, “And then..”

Line 54-55: This sentence sounds incomplete

Line 57-58: processing process?? Restructure this sentence

Line 62: more and more?? This doesn’t sound grammatically correct. Please re-structure this sentence

Line 62-67: Restructure this statement, please. Add a reference. You tend to use the word, “then” excessively. Try writing without using the word, “then”

Line 70-72: This sentence does make sense. Revise this sentence, please. 

The introduction could need some improvement. Please take some time to revise it.

Line 91-92: The statement is not clear, it needs revision.

Line 97: Freed drinking water? What are you trying to mean with this statement? Did birds have ad-lib access to drinking water throughout the experiment?? 

Line 121: Did you collect blood vessels?? Which blood vessels did you collect? How did you collect blood vessels? Did you use a specific procedure to collect the blood vessels? Be specific and clear on how this was done and provide a reference in case it exists.

Line 157-158: You indicate that cecal samples were sent to the lab. You don’t state why you took the cecal samples. Did you extract DNA from the cecal content? If so, that is bacterial DNA from the cecal samples. Could you please revise this section and clearly state the rational for sampling those cecal samples.

Line 225: The phenotype data for IMF, AF, TG, TC….

Figure 5: Increase the font size for all the text (including axes) for figures A-E. The figures should be readable.

Line 280-283: Correct the grammar. You are reporting results, hence, this should be written in an appropriate form.

Figure 6: Increase the text font size.

Line 312: tended? I don’t think you need to use this word

Line 406-408: You indicated that, “compound probiotics significantly reduced TG and increased HDL-C…………., and reduced the occurrence of cardiovascular disease”. I understand the health benefits of reduced TG and increased HDL, however, you didn’t observe the reduction of cardiovascular disease in the current experiment. I think you could improve your conclusion. 

Author Response

Dear Reviewer 2:

On behalf of my co-authors, we thank you very much for giving us an opportunity to revise our manuscript, we appreciate editor and reviewers very much for their positive and constructive comments and suggestions on our manuscript entitled “animals-1873223”. 
We carefully studied the comments made by the reviewers and revised them. We provide a revised version for your review. In addition, we recruited native English speakers to revise our articles. Attached please find the reply to Reviewer  2, which we would like to submit for your kind consideration. 
We would like to express our great appreciation to you and reviewers for comments on our paper. Looking forward to hearing from you.
Thank you and best regards.

Yours sincerely,
Mrs. Zhang
Corresponding author:
Name: Lihuan Zhang
E-mail: [email protected]

Round 2

Reviewer 1 Report

The manuscript entitled "WGCNA analysis of important modules and hub genes of compound probiotics regulating lipid metabolism in heat-stressed broilers" have been sufficient revised. The authors have revised all important issues of concern to us, therfore, we recommend acceptance for publication in present version to our journal.

Reviewer 2 Report

Thank you for addressing my comments and suggestions!